# AGC family kinase 1 participates in trogocytosis but not in phagocytosis in *Entamoeba histolytica*

Somlata[1], Kumiko Nakada-Tsukui[2] & Tomoyoshi Nozaki[2,3,4]

The protozoan parasite *Entamoeba histolytica* is the aetiologic agent of amoebiasis, an endemic infection in developing countries with considerable morbidity and mortality. Recently, trogocytosis has been recognized as the key step in amoebic cytolysis and invasion, a paradigm shift in understanding pathogenicity of this organism. Here we report that AGC family kinase 1 is specifically involved in trogocytosis of live human cells and does not participate in phagocytosis of dead cells. Live imaging reveals localization of this kinase in the long and thin tunnels formed during trogocytosis but not in the trogosomes (endosomes formed after trogocytosis). Silencing of the specific gene leads to a defect in CHO cell destruction and trogocytosis while other endocytic processes remain unaffected. The results suggest that the trogocytic pathway is likely to be different from phagocytosis though many of the steps and molecules involved may be common.

---

[1] School of Life Sciences, Jawaharlal Nehru University, New Delhi 110067, India. [2] Department of Parasitology, National Institute of Infectious Diseases, 1-23-1 Toyama, Shinjuku-ku, Tokyo 162-8640, Japan. [3] Graduate School of Life and Environmental Sciences, University of Tsukuba, 1-1-1 Tennodai, Tsukuba, Ibaraki 305-572, Japan. [4] Graduate School of Medicine, The University of Tokyo, 7-3-1 Hongo, Tokyo 113-0033, Japan. Correspondence and requests for materials should be addressed to Somlata (email: somlata83@gmail.com) or to T.N. (email: nozaki@m.u-tokyo.ac.jp)

moebiasis is a major cause of morbidity and mortality worldwide infecting as many as 50 million people every year[1]. The majority of infected people do not show clinical symptoms and are asymptomatic. Among the fraction of the people who show clinical symptoms, invasive disease in the colon is more common than in the liver. It is unclear which host-pathogen factors prompt tissue invasion by the parasite and why only a few patients show extraintestinal infection. Invasive amoeba can lead to severe tissue destruction ranging from intestinal ulceration, causing diarrhoea, colitis and dysentery, to abscess formation at other sites such as liver and lungs[2–4]. Pathogenesis in amoebiasis is due to invasion of intestinal epithelial tissues or extraintestinal tissues through blood vessels by trophozoites. Tissue invasion involves three steps, adherence, cytolysis and phagocytosis. Though we understand some of the pathways that are involved in these processes, detailed mechanisms are not clear. Understanding these mechanisms will help us to design better prophylactic and therapeutic measures.

A number of molecules, such as Gal/GalNAc lectin, serine-rich *Entamoeba histolytica* proteins and lipopeptidephosphoglycan, have been suggested as possible molecules involved in attachment of *E. histolytica* to epithelial cells. Antibodies against some of these molecules have been shown to block adherence and cytolysis[5]. Cytopathic properties of amoeba have been suggested to be due to pore forming factors, such as amoebapore, and some of the secreted hydrolytic enzymes, cysteine proteases being the major one[6]. The prevalent belief was that amoeba first kills the host cells in a contact-dependent manner and then phagocytosis of dead cells takes place and this helps cells to invade[7, 8]. But recently, a new model of cell killing and invasion has been established. In this alternate model, host-cell destruction is thought to be through the process of trogocytosis, that is, ingestion of several fragments of live cells in multiple steps leading to elevation of $Ca^{2+}$ in cytosol, loss of plasma membrane integrity

and eventually death of the cells[9]. Only live cells undergo trogocytosis by *E. histolytica*, whereas dead cells are taken up by phagocytosis[9]. It appears that amoebic trogocytosis overlaps with the signalling pathway of erythrophagocytosis involving EhC2PK and PI3K, suggesting that the same pathway is at least partially used for different endocytic processes. However, it is still not clear what makes only live cells undergo trogocytosis.

Most endocytic processes, such as macropinocytosis, phagocytosis and trogocytosis, depend on phosphoinositides and the actin cytoskeleton. The signalling events coupling these two components are incompletely understood in *E. histolytica*. EhCaBP1, EhCoactosin, EhAK and EhC2PK have been shown to be involved in initiation of phagocytosis and progression of phagocytic cups in *E. histolytica*[10–13]. These molecules are thought to be recruited at the particle attachment site and initiate a process that leads to actin dynamics[12]. Myosin1B and EhCaBP3 have been implicated in phagosome fusion and separation from plasma membrane[14, 15]. Though some of these molecules are also involved in fluid-phase pinocytosis and trogocytosis, it is not clear how these three processes (phagocytosis, pinocytosis and trogocytosis) differ from each other at the molecular level.

PtdIns(3,4,5)P₃ is an important molecule involved in regulation of dynamic processes like motility and endocytosis. In mammalian cells, the class I PI3K-mediated generation of PtdIns(3,4,5)P₃ during phagocytosis helps in recruitment of RhoGEFs, Vav1 and ARF, leading to assembly of protein complexes required for actin remodelling during engulfment of particles[16, 17]. Moreover, Vav and Rac proteins are phosphorylated in a PtdIns(3,4,5)P₃-dependent manner, resulting in activation of actin machinery through Wiskott-Aldrich syndrome protein and Arp2/3 pathway[18].

Given the importance of PtdIns(3,4,5)P₃ in mammalian cells, we here hypothesized that PtdIns(3,4,5)P₃-binding proteins are involved in various endocytic processes and may shed light on

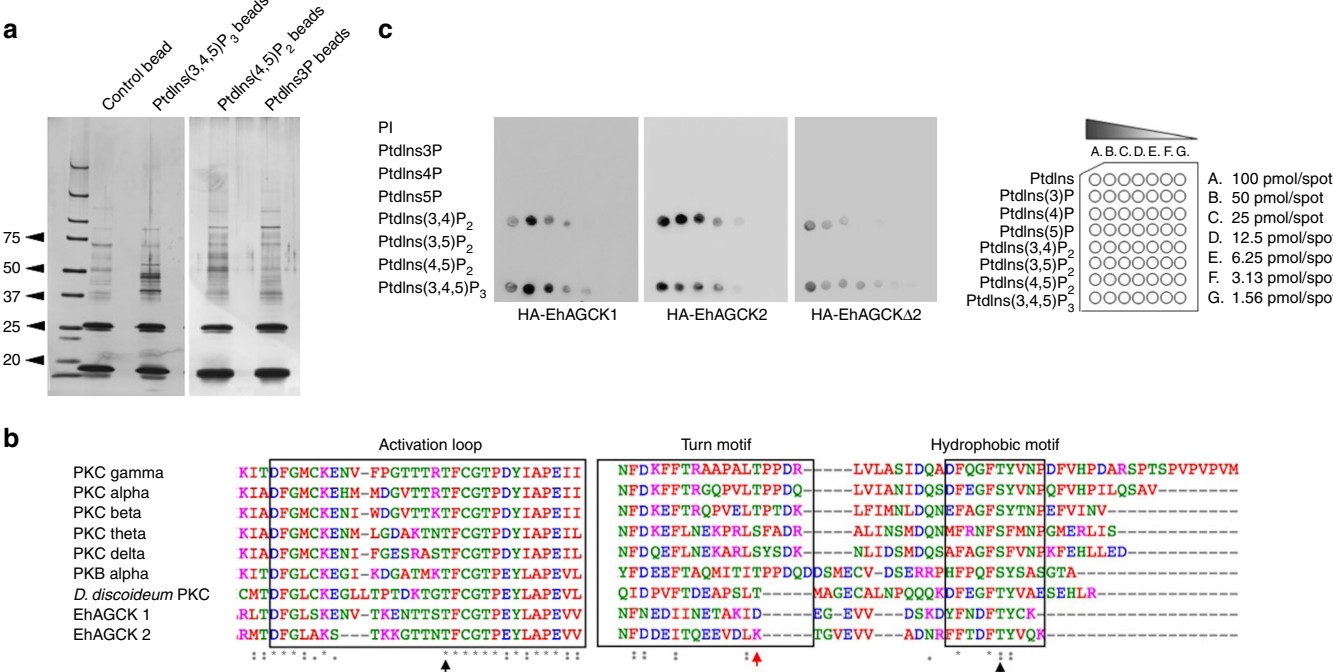

**Fig. 1** Identification of PtdIns(3,4,5)P₃-binding proteins. **a** Proteins eluted from PtdIns(3,4,5)P₃, PtdIns(4,5)P₂, PtdIns3P and lipid-free control beads were resolved on SDS-PAGE gel followed by silver staining. **b** Lipid overlay assay with HA-tagged proteins expressed in *E. histolytica* trophozoites. **c** Multiple sequence alignment with various PKC isoforms from human and protein kinase from *D. discoideum* showing important conserved phosphorylation sites in EhAGCK1 and EhAGCK2 activation loop and hydrophobic motif (marked by *black arrow*). The conserved phosphorylation site is absent in the turn motif of both proteins (marked by *red arrow*)

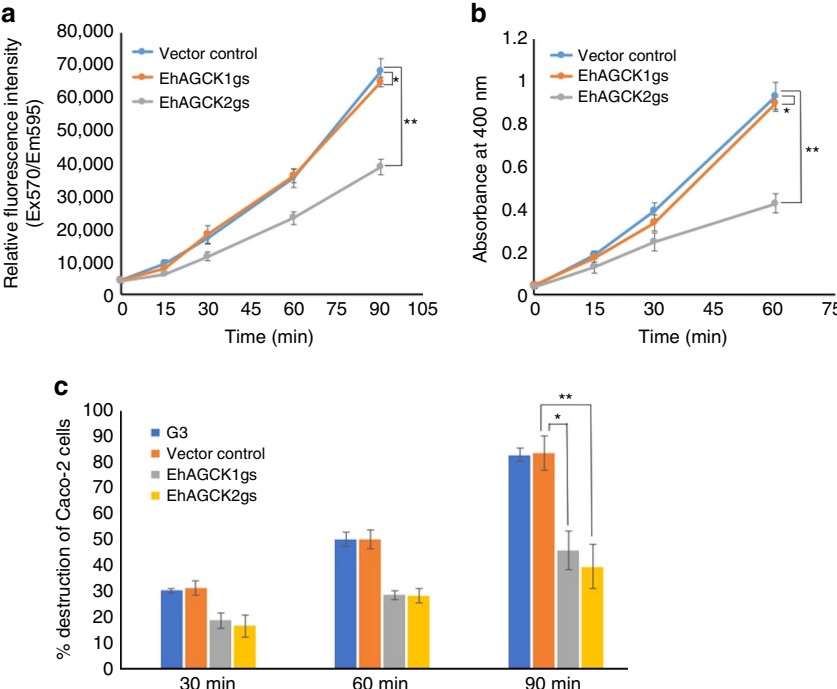

**Fig. 2** Effect of EhAGCK1 and EhAGCK2 gene silencing on *E. histolytica* trophozoites. **a** Trophozoites show defect in pinocytosis of RITC dextran upon EhAGCK2 silencing in comparison to vector control, as measured by RITC dextran uptake by trophozoites. The experiment was repeated three times independently in triplicates ($N = 3$ with *error bars* indicating the standard deviation). The statistical comparisons were made by Student's *t*-test for the final time point, the *P* value for * is 0.388 and for ** is 0.0003. **b** Quantitation of erythrophagocytosis of gene-silenced trophozoites was performed with cell lines as indicated by erythrophagocytosis assay. The experiment was repeated three times independently in triplicates ($N = 3$ with *error bars* indicating the standard deviation). The statistical comparisons were made by *t*-test for the final time point, the *P* value for * is 0.367 and **<0.0001. **c** Trophozoites silenced for EhAGCK1 and EhAGCK2 showed defect in Caco-2 cells monolayer destruction in comparison to vector control in time-dependent manner. The experiment was repeated three times independently in duplicates ($N = 3$ with *error bars* indicating standard deviation). The statistical comparisons were made by unpaired *t*-test, with *P* values for * and ** are 0.003 and 0.0005, respectively

molecular differences between phagocytosis and trogocytosis in *E. histolytica*. We identified 18 PtdIns(3,4,5)P₃-binding proteins, all of which contained a PH domain. Furthermore, we showed that one AGC family kinase, EhAGCK1, is specifically involved in trogocytosis but not in phagocytosis of dead cells, while EhAGCK2 is involved in all actin-dependent endocytic processes. This study identifies an AGC family kinase that is specifically involved in amoebic trogocytosis, shedding light on the signalling pathways that distinguish trogocytosis and phagocytosis.

## Results

**Identification of PtdInsP3-binding proteins**. For identification of PtdIns(3,4,5)P₃-interacting proteins by affinity purification, we used PtdIns(3,4,5)P₃-conjugated beads. The amoebic lysate was ultracentrifuged and supernatant obtained was used for incubation with PtdIns(3,4,5)P₃-conjugated beads. Beads conjugated with PtdIns(4,5)P₂ and PtdIns(3)P were used as control and beads with no lipid conjugates were also included in the experiments. The SDS-PAGE gel profile showed distinct patterns of PtdIns(3,4,5)P₃-binding proteins as compared to the control experiment (Fig. 1a). The proteins were then identified by mass spectrometry. The mass spectrometry results revealed 18 proteins that bound uniquely to PtdIns(3,4,5)P₃ resin (Supplementary Table 1). Only epsin N-terminal homology (ENTH) domain-containing protein was common between PtdIns(3,4,5)P₃- and PtdIns(4,5)P₂-binding proteins. Mass spectrometry data revealed the presence of different proteins, in case of PtdIns(3)P, which included some EF-hand containing proteins. The presence of different proteins in all PI-coupled resin indicated that the

profiling was specific for the three forms of PI included in the experiment.

The identified proteins were screened for their role in biological processes as described in Supplementary Fig. 1. On the basis of microscopy data, the selected proteins were further assessed for their role in pinocytosis, erythrophagocytosis, motility and destruction of live CHO cells. After screening and assigning roles in biological processes, we found an AGC family kinase (EhAGCK1) which had the ability to bind PtdIns(3,4,5)P₃ and a specific role in amoebic trogocytosis. We also included another AGC kinase found in the mass spectrometry result for comparative analysis (EhAGCK2).

**Domain organization of EhAGCK1 and EhAGCK2 and in silico analysis**. The two kinases selected for the study belong to the AGC family and have a very similar domain organization comprising of a PH domain at the N terminus, a kinase domain and a conserved C-terminal hydrophobic region[19]. The multiple alignment of kinase domains with homologues in human and *Dictyostelium discoideum* revealed conserved motifs and residues necessary for kinase activity (Supplementary Fig. 2). The AGC family kinases are regulated through ordered phosphorylation of three residues. Figure 1b shows three conserved phosphorylation sites known in AGC family kinases, but in case of EhAGCK1 and EhAGCK2 only two of the phosphorylation sites could be identified in silico (Fig. 1b). The phosphorylation site in the activation loop, which is in proximity to the catalytic site (T500 in case of PKC alpha), is essential for activation of the kinase[19]. Its phosphorylation results in subsequent phosphorylation of the so-called "turn motif" in the proline-rich domain[19]. Once the

kinase has been activated, only the phosphorylation at the turn motif is required for catalytic activity. EhAGCK1 and EhAGCK2 lack this conserved phosphorylation site in the turn motif as shown in the sequence alignment (Fig. 1b). The phosphorylation site in the C-terminal hydrophobic motif is conserved in both, EhAGCK1 and EhAGCK2, even though it is a threonine instead of the more common serine. The hydrophobic sequences near the phosphorylation site form a docking site for PDK-1 (phosphoinositide-dependent kinase 1). This site is available for binding to PDK-1 in the unphosphorylated enzyme but gets masked once PDK-1 phosphorylates the enzyme. Overall, EhAGCK1 and EhAGCK2 contain the majority of the conserved features of AGC family kinases. However, it is possible that the regulation of these kinases differs in *Entamoeba* and experimental characterization of the phosphorylation sites and their role in regulation of enzyme activities will be necessary.

**Lipid-binding specificity of EhAGCK1 and EhAGCK2**. Both EhAGCK1 and EhAGCK2 were expressed as haemagglutinin (HA)-tagged proteins in *E. histolytica* trophozoites to analyse their role in cell biology (Supplementary Fig. 3a). We created a cell line expressing only the PH domain of the protein (HA-EhAGCKΔ2) to analyse lipid-binding specificity of the domain (Supplementary Fig. 3b). The HA-tagged proteins were checked for lipid binding by lipid overlay assay (Fig. 1c). The results showed that proteins were able to bind $PtdIns(3,4,5)P_3$ on lipid array with some cross reactivity towards $PtdIns(3,4)P_2$. However, no binding was detected for $PtdIns(4,5)P_2$ and PtdIns3P. This kind of cross reactivity towards $PtdIns(3,4)P_2$ by PH domain of AGC kinases is well known and documented[20].

**E. histolytica cells with silenced EhAGCK1 or EhAGCK2 expression**. In order to analyse the biological function of EhAGCKs in various processes of *E. histolytica*, we silenced the expression of the genes by antisense small RNA-mediated transcriptional gene silencing in the G3 strain[21, 22]. The specific gene repression was confirmed by reverse transcription PCR of corresponding cDNA. Complete silencing of the specific genes was achieved in the trophozoites in comparison to the vector control (Supplementary Fig. 4). Further, expression of EhAGCK2 in EhAGCK1 gene-silenced cells (EhAGCK1gs) and vice versa was checked, and the results showed specific silencing of the respective genes.

**Effects of gene silencing on pinocytosis**. Amoebic cells in culture and also in host gut derive most of the nutrients through pinocytosis or macropinocytosis. Pinocytosis by *E. histolytica* trophozoites was assayed by RITC dextran uptake in a time-dependent manner. The cells silenced for EhAGCK1 expression did not exhibit any defect in pinocytosis as compared to vector control, while EhAGCK2-silenced cells showed ~$42 \pm 5$% ($N = 3$, $\pm$ represents standard deviation) decrease in pinocytosis at 90 min. This indicated involvement of EhAGCK2 in pinocytosis (Fig. 2a). These results were further supported by live cell imaging experiments (see below). Overexpression of HA-tagged wild-type proteins did not lead to any increase in pinocytosis by the trophozoites (Supplementary Fig. 5a).

**Effects of gene silencing on erythrophagocytosis**. Erythrophagocytosis is an important property and a marker of virulence in *E. histolytica*. To analyse the role of EhAGCK1 and EhAGCK2 in erythrophagocytosis, we again utilized the trophozoites silenced for the respective genes. The performed assay measures the amount of haem ingested by the parasite in a time-dependent manner[13]. The results showed that trophozoites

silenced for EhAGCK1 did not show any defect in red blood cell (RBC) uptake, while trophozoites silenced for EhAGCK2 had a defect of about $54 \pm 4.6$% ($N = 3$) for erythrophagocytosis in comparison to vector control (Fig. 2b). This showed that EhAGCK2 is involved in erythrophagocytosis. We also analysed the effect of overexpression of HA-tagged EhAGCK1 and EhAGCK2 in trophozoites, which did not lead to any increase in erythrophagocytosis rates (Supplementary Fig. 5b).

**Effect of gene silencing on destruction of live cells**. *E. histolytica* cells ingest live mammalian cells primarily by trogocytosis. This kind of endocytosis becomes important during the invasive phase of the parasite. Destruction of monolayer of CHO and Caco-2 cells by trophozoites mimics the destruction of host cells by the parasite during invasion. Caco-2, an epithelial colorectal adeno-carcinoma cell line, provides a close resemblance to gut epithelium during destruction by trophozoites. In order to reveal the role of EhAGCK1 and EhAGCK2 in this process, we labelled CHO and Caco-2 cells by CellTracker blue and then incubated them with indicated amoeba cell lines[23]. Destruction is inversely related to fluorescence of intact cells. We analysed destruction of Caco-2 cells by gene-silenced trophozoites in a time-dependent manner (Fig. 2c). In case of vector control, about $83.57 \pm 6.6$% ($N = 3$) cells were destroyed by the trophozoites in 90 min while EhAGCK1gs amoebic cells destroyed $45.89 \pm 7.5$% ($N = 3$) of target cells and EhAGCK2gs trophozoites destroyed $39.6 \pm 8.5$% ($N = 3$) of Caco-2 cells in the same time frame. In case of CHO cells, almost $70 \pm 1.8$% ($N = 3$) cells were destroyed by the control trophozoites in 1 h while this destruction was only $33 \pm 3.9$% ($N = 3$) in case of EhAGCK1-silenced strain and $26 \pm 3.2$% ($N = 3$) for EhAGCK2-silenced cells (Supplementary Fig. 6a). This showed that EhAGCK1 is involved in destruction of live host cells specifically and does not play any role in ingestion of RBCs and pinocytosis, while EhAGCK2 may be a common protein in actin-dependent endocytic processes in amoeba.

**Dominant-negative phenotype by kinase dead mutants**. Constitutive overexpression of kinase dead mutants of EhAGCK1 (HA-EhAGCK1mut) and EhAGCK2 (HA-EhAGCK2mut) resulted in a dominant-negative phenotype, that is, a defect in Caco-2 and CHO cell monolayer destruction. The HA-EhAGCK1mut overexpressing amoebic cells could destroy only $44.2 \pm 3.7$% ($N = 3$) of Caco-2 cells in comparison to $81.4 \pm 7.6$% ($N = 3$) by vector control trophozites (Supplementary Fig. 6b). Similar defect was also detected in EhAGCK2mut overexpressing amoebic cells where the destruction was $45.3 \pm 5.12$% ($N = 3$) (Supplementary Fig. 6b). We also analysed the dominant-negative effect on CHO cell destruction. The control trophozoites destroyed $76 \pm 2.4$% ($N = 3$) of CHO cells in an hour, whereas trophozoites overexpressing HA-EhAGCK1mut could destroy $43 \pm 3.3$% ($N = 3$) of CHO cells. The destruction was $42.8 \pm 2.8$% ($N = 3$) in case of trophozoites overexpressing HA-EhAGCK2mut in comparison to vector control (Supplementary Fig. 6c). This experiment confirmed the role of EhAGCK1 and its kinase activity in destruction of live cells.

Since motility of the parasite is directly related to virulence and cell destruction, we investigated the effect of gene silencing on motility by transwell migration assay. In this assay, the upper and lower chambers are separated by porcine mucin and bovine serum is contained only in the lower chamber. The trophozoites silenced for EhAGCK1 showed ~20% defect in motility across the barrier towards serum containing medium in the lower chamber as compared to vector control (Supplementary Fig. 6d). One should note that under experimental conditions of a confluent monolayer of mammalian cells, as used in our destruction assays,

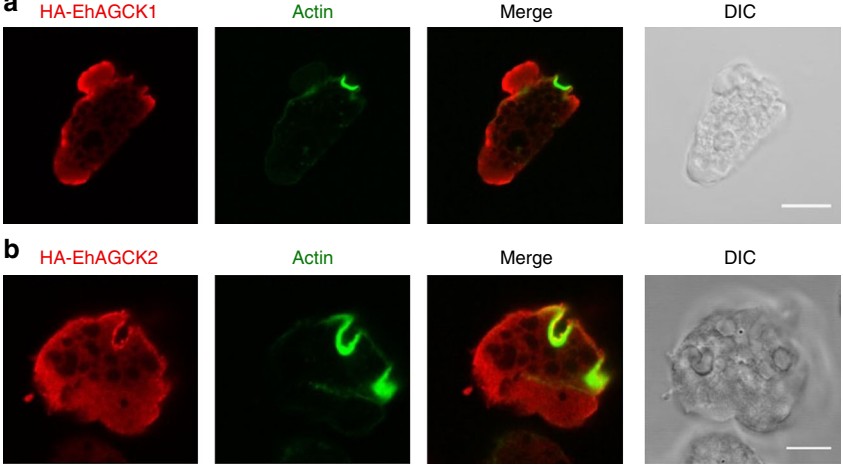

**Fig. 3** Immunolocalization of N-terminal HA-tagged protein in fixed trophozoites. **a** HA-tagged EhAGCK1 and **b** HA-tagged EhAGCK2-expressing trophozoites were fixed and stained for tagged protein and actin. The cells were double labelled with Alexa 546 (EhAGCK1 or EhAGCK2) and FITC-phalloidin (actin). (*Scale bar*, 10 μm; *DIC* differential interference contrast)

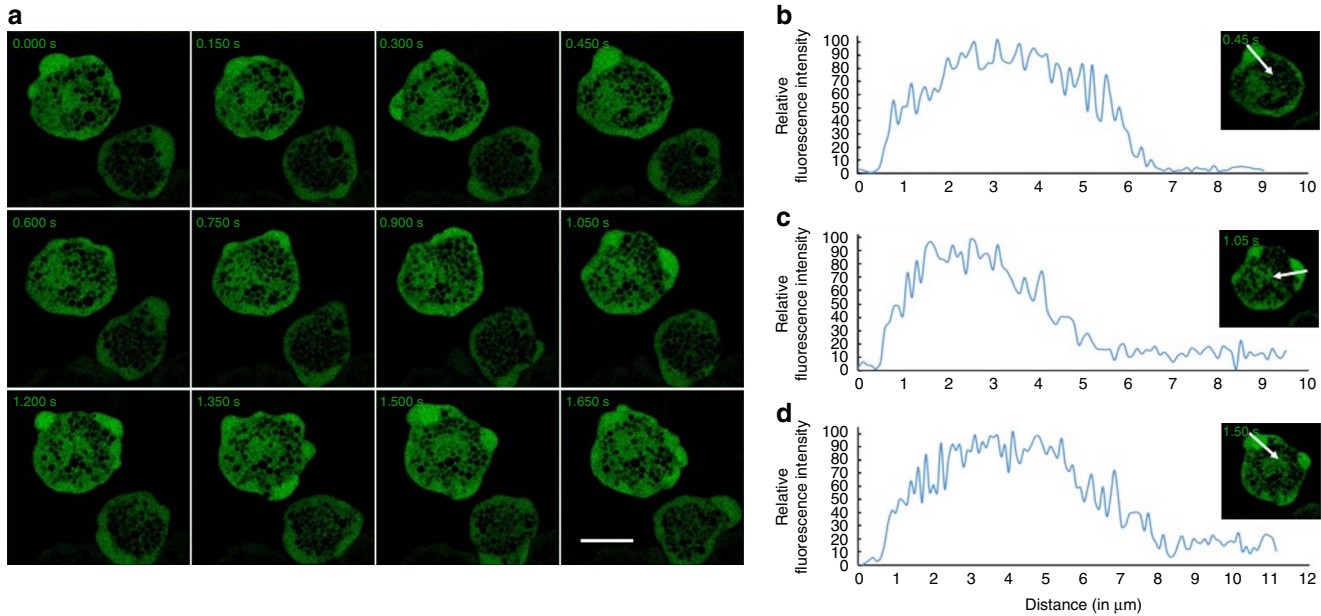

**Fig. 4** Live imaging montage showing localization of GFP-EhAGCK1 in normal motile trophozoites. **a** Montage showing a time series of motile trophozoites expressing GFP-EhAGCK1. A number of pseudopods in different directions can be visualized. **b–d** The pseudopods in three different time frames have been analysed for GFP-EhAGCK1 intensity along the marked *arrow line*. The graph shows intensity of GFP-EhAGCK1 in pseudopods vs. cytoplasm and plasma membrane along the *line*. (*Scale bar*, 10 μm)

parasite motility is likely of limited advantage for destruction. Hence, the defect in motility is unlikely to explain the observed defect in the destruction experiment. However, the possibility that EhAGCK1 along with other factors plays a role in directional motility of parasites in the gut environment of the host remains and needs to be further investigated in future experiments.

**Effect of gene silencing on Caco-2 cell trogocytosis.** The internalization of fragments of live target cells is a hallmark of amoebic trogocytosis. To physically quantify the defect in amoebic trogocytosis, we utilized microscopy with gene-silenced cells. The amoebic cells were incubated with labelled live Caco-2 cells and fixed for microscopy. Forty-five cells in each transfected amoebic cell line were randomly selected in each independent

experiment and z-sectioned, and the reconstructed images were analysed by IMARIS 7.6 software. We measured the average total intensity of labelled Caco-2 fragments taken up by the tropho-zoites and found that vector control showed significantly higher uptake of live Caco-2 cells as compared to EhAGCK1- and EhAGCK2-silenced cells in three independent experiments (Supplementary Fig. 7a). The representative reconstructed images used for analysis are shown in Supplementary Fig. 7b, c, d. This observation confirmed the defect in uptake of live cells due to silencing of EhAGCK1 and EhAGCK2.

**Immunofluorescence of HA-tagged EhAGCK1 and EhAGCK2.** In order to understand the relation between cellular localization and function of EhAGCK1 and EhAGCK2 in trophozoites,

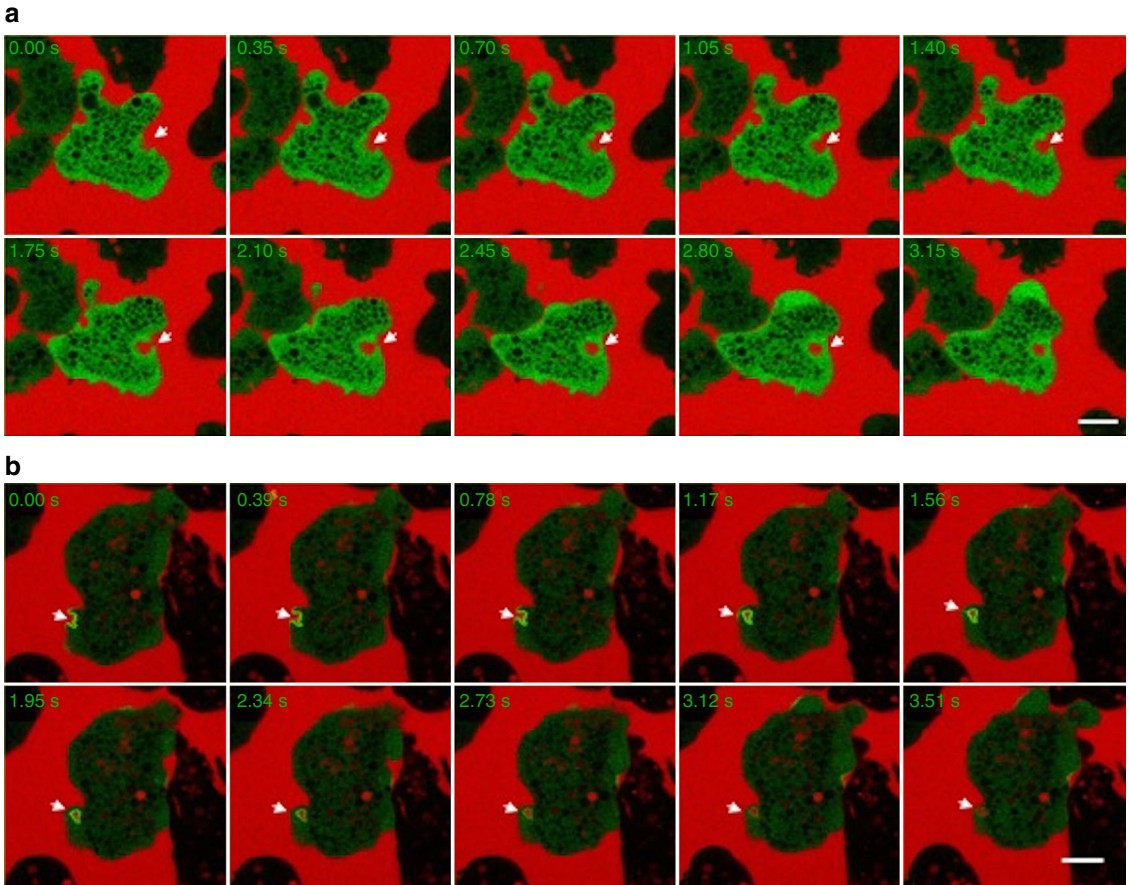

**Fig. 5** Localization of GFP-tagged EhAGCK1 and EhAGCK2 during pinocytosis. Time series montage showing localization of **a** GFP-EhAGCK1 and **b** GFP-EhAGCK2 during pinocytosis of FITC dextran by amoebic trophozoites (the site of pinocytosis is marked with *arrow*; *Scale bar*, 10 μm)

HA-tagged proteins were analysed after fixing. EhAGCK1 was localized mainly near the plasma membrane and was associated with pseudopods (Fig. 3a). EhAGCK1 did not co-localize with actin-positive endocytic structures. EhAGCK2 was present homogenously in the cytoplasm and close to the plasma membrane. Furthermore, EhAGCK2 co-localized with actin-rich endocytic structures (Fig. 3b). The data suggests that, in contrast to EhAGCK2, EhAGCK1 is not associated with endocytic structures in resting trophozoites and hence not involved in pinocytosis, supporting our measurement of pinocytosis by RITC dextran uptake. As expected, HA-EhAGCK1mut and HA-EhAGCK2mut localized in a similar manner as wild-type protein (Supplementary Fig. 8a, b). This indicates that an active kinase domain is not necessary for subcellular localization of the proteins, and instead the PH domain likely determines localization (Supplementary Fig. 8a, b). Furthermore, to rule out a possible influence of the HA-tag on localization of EhAGCK1 and EhAGCK2, we analysed localization of a cytoplasmic HA-tagged protein, BAR domain-containing protein, under the same conditions (Supplementary Fig. 8c). HA vector control transfected trophozoites served as negative control (Supplementary Fig. 8d). The HA-tagged control protein neither localized to phagocytic cups nor to phagosomes, showing that the HA-tag does not affect localization of EhAGCK1 or EhAGCK2 to endocytic structures.

**Live cell imaging of GFP-tagged EhAGCK1 and EhAGCK2.** The trophozoites were transfected to express N-terminal GFP-tagged full-length wild-type protein in a constitutive expression system (Supplementary Fig. 3c). Live cell imaging of GFP-EhAGCK1 expressing normal trophozoites showed its localization to pseudopod-like structures (Fig. 4a and Supplementary Movie 1). We analysed the intensity of GFP-EhAGCK1 along the arrow shown in the micrograph at different time points. The intensity profile of three different pseudopods formed by the trophozoite showed enrichment along the line of analysis (Fig. 4b–d). In case of GFP-EhAGCK2, the protein localized uniformly in the cytosol, with enrichment at certain sites on the plasma membrane (Supplementary Fig. 9a). A similar GFP fluorescence intensity analysis was performed for GFP-EhAGCK2 and the intensity plot along the line showed insignificant enrichment of the protein in pseudopods (Supplementary Fig. 9b–d).

Next, we analysed localization of GFP-EhAGCK1 and GFP-EhAGCK2 during pinocytosis of RITC dextran. GFP-EhAGCK1 showed no preferential enrichment at the site of pinocytosis (Fig. 5a). This data further supported our previous results from pinocytosis assay, confirming that EhAGCK1 does not play a role in the process. In contrast, GFP-EhAGCK2 was associated with pinocytic vesicles during RITC dextran uptake. After closure of the pinocytic vesicle GFP-EhAGCK2 dissociated rapidly (Fig. 5b).

We analysed erythrophagocytosis using time-lapse live imaging by feeding labelled human RBC to trophozoites expressing GFP-EhAGCK1. Even though it has been reported that RBCs can undergo trogocytosis[9], we were unable to clearly detect trogocytosis by microscopy in these experiments. It may be that the small and disc-like RBCs are preferentially taken up by phagocytosis while the flat and large, live host cells are preferably ingested through trogocytosis. At the site of erythrophagocytosis,

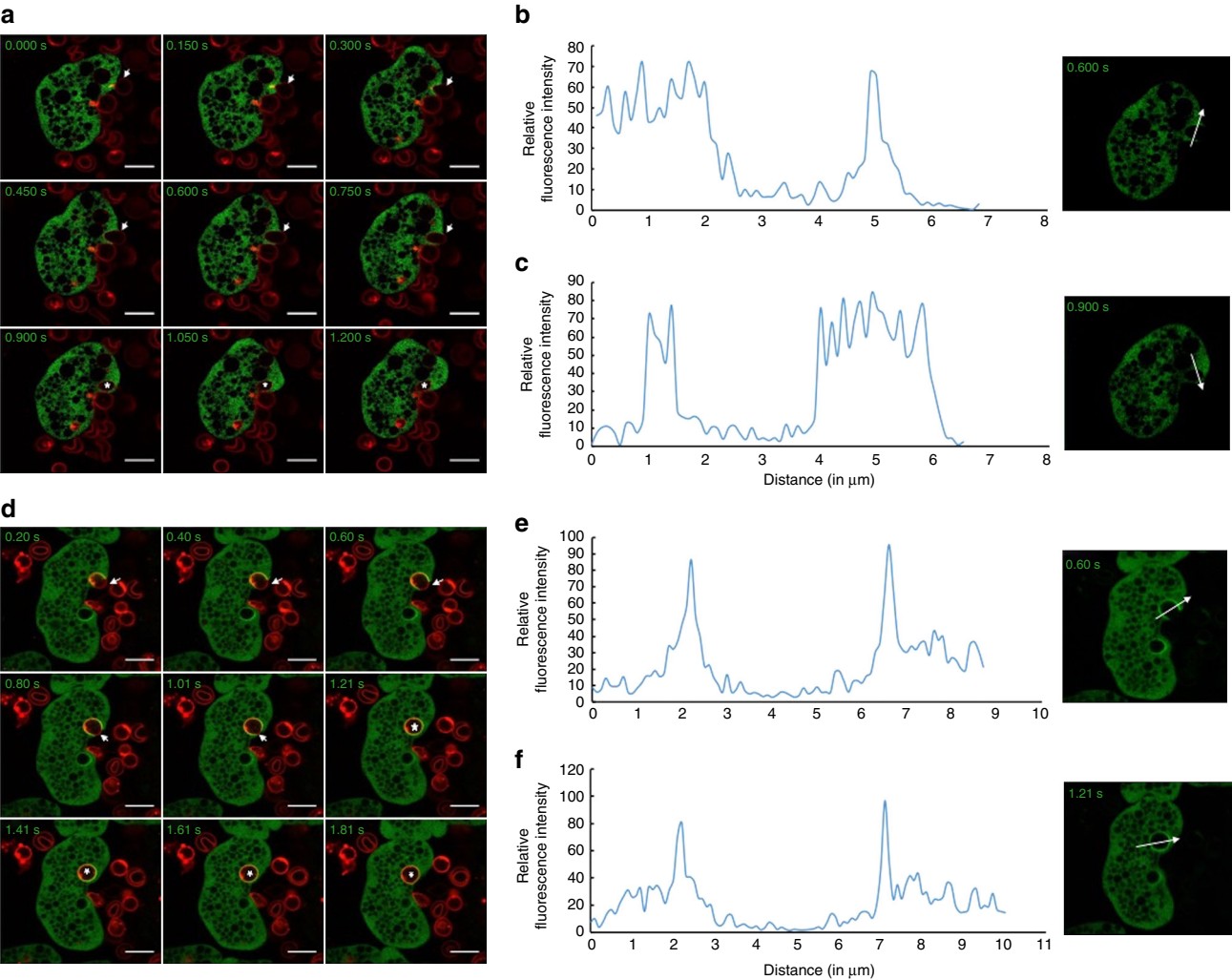

**Fig. 6** Time series montage showing localization of GFP-EhAGCK1 and EhAGCK2 during erythrophagocytosis. **a** The montage of trophozoite expressing GFP-EhAGCK1 engulfing labelled human RBC. **b**, **c** Analysis of intensity of GFP-EhAGCK1 along the *line* marked across the phagocytic cup **b** and newly formed phagosome **c**. **d** Montage of a time series of GFP-EhAGCK2 expressing trophozoite undergoing erythrophagocytosis. **e**, **f** The intensity analysis of GFP-EhAGCK2 along the *line* drawn reveals its enrichment in the membrane of phagocytic cups **e** and newly formed phagosome **f**. Phagocytic cups are marked by *arrow* and internalized RBCs in new phagosome are marked by *asterisk*. (*Scale bar*, 10 μm)

we could not detect specific enrichment of GFP-EhAGCK1 (Supplementary Movie 2). The montage shows no preferential localization of GFP-EhAGCK1 at any stage of the process (Fig. 6a and Supplementary Fig. 10a, b). This observation was supported by quantitatively analysing the micrographs at two time points, which showed no significant change in intensity of GFP-EhAGCK1 at the site of erythrophagocytosis (Fig. 6b, c). In contrast, GFP-EhAGCK2 localized to the erythrophagocytic cups and associated with the newly formed phagosome (Fig. 6d, Supplementary Movie 3 and Supplementary Fig. 11a, b) but disassociated very rapidly soon after (about 0.4–0.8 s). The quantitation of fluorescence intensity of GFP-EhAGCK2 along the arrow shown on micrographs consistently revealed an increase in the membrane in contact with RBC as compared to other regions (Fig. 6e, f).

Trogocytosis of live CHO cells by *E. histolytica* trophozoites was analysed using live cell imaging and we used these images for comparing the localization of GFP-EhAGCK1 and GFP-EhAGCK2 during the process. For describing the structures formed after trogocytosis, we will use the term "trogosomes". The imaging data revealed recruitment of GFP-EhAGCK1 to narrow tunnels formed during the ingestion of live CHO cells (Fig. 7a and

Supplementary Movie 4). The protein preferentially localized to tunnels and dissociated rapidly from newly formed trogosomes. Our analysis of intensity of GFP-EhAGCK1 along the arrow drawn across the tunnel at different time intervals showed enrichment of GFP-EhAGCK1 to the tunnel membrane, while the rest of the plasma membrane and cytoplasm showed low levels of GFP-EhAGCK1 (Fig. 7b). Quantitative measurements also showed that the membrane around the newly formed trogosome is not enriched for GFP-EhAGCK1 as compared to plasma membrane and cytosol, indicating its rapid disassociation (Fig. 7c). We observed this pattern consistently in independent experiments (Supplementary Fig. 12a, b). Time-lapse montage of GFP-EhAGCK2-expressing cells showed that GFP-EhAGCK2 is recruited to the site where the trophozoite makes contact with the live cell and then follows through the tunnel (Fig. 7d, Supplementary Movie 5 and Supplementary Fig. 13a, b). Quantitative measurement of fluorescence intensity along the arrow shown on micrographs showed enrichment of GFP-EhAGCK2 on the membrane in contact with live CHO cells as well as in the tunnel formed during trogocytosis (Fig. 7e). GFP-EhAGCK2 was also absent in newly formed trogosomes (Fig. 7f). Furthermore, we analysed the localization of

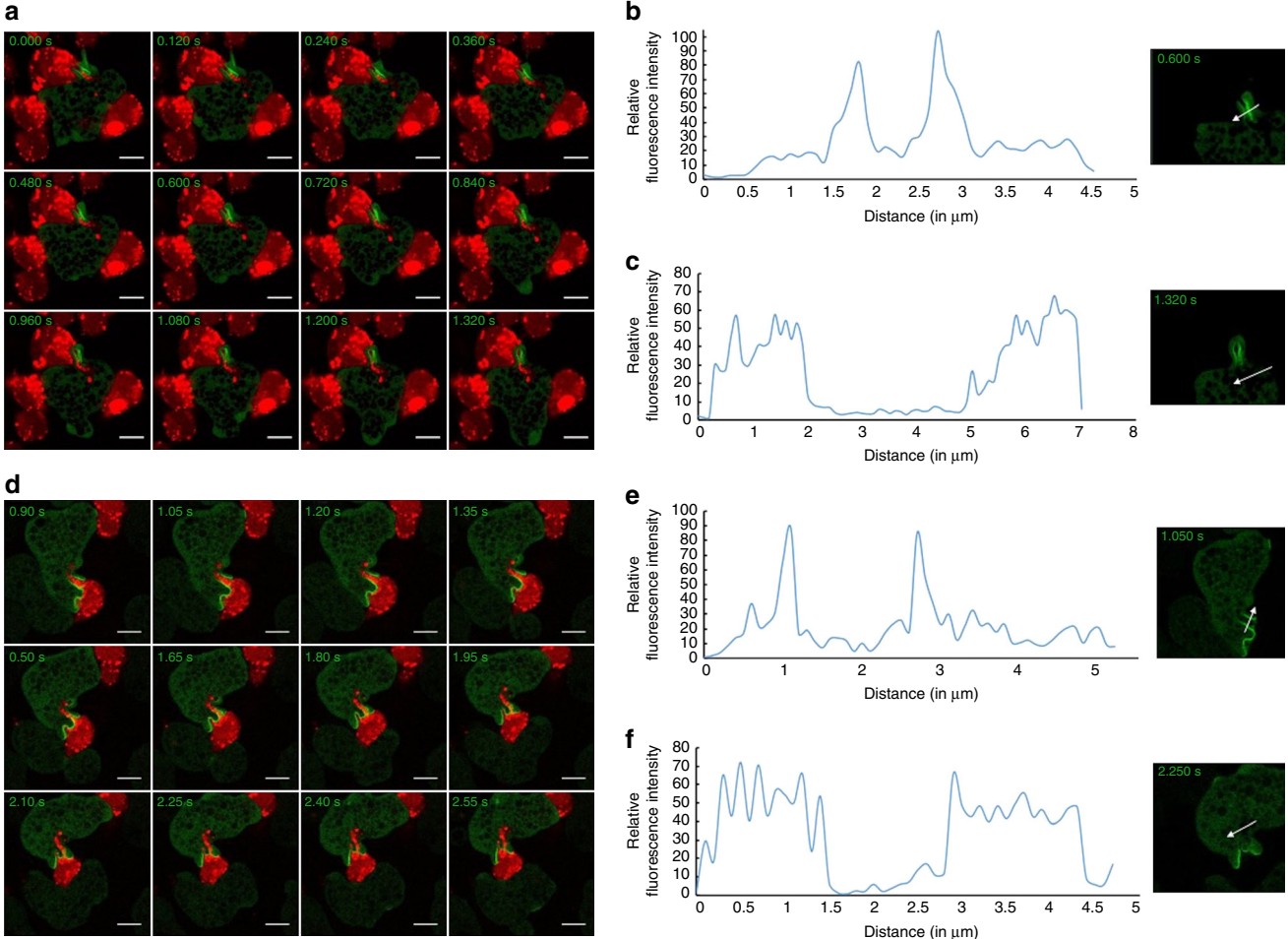

**Fig. 7** Montage showing recruitment of GFP-EhAGCK1 and -EhAGCK2 at the site of trogocytosis. **a** Montage of live imaging of GFP-EhAGCK1 expressing trophozoite ingesting labelled live CHO cells by trogocytosis. **b** The *plot* showing intensity of GFP-EhAGCK1 across the tunnel formed during amoebic trogocytosis along the *line* drawn. **c** GFP-EhAGCK1 intensity across the newly formed trogosome along the *line* drawn through. **d** Time series montage of trophozoite-expressing GFP-EhAGCK2 engulfing labelled live CHO cells by trogocytosis. **e** Analysis of GFP-EhAGCK2 intensity across tunnel formed during trogocytosis of live CHO cells along the *line*. **f** Plot showing GFP-EhAGCK2 intensity along the *line* drawn across newly formed phagosome. (*Scale bar*, 10 μm)

GFP-EhAGCK1 and GFP-EhAGCK2 during trogocytosis of live Caco-2 cells. GFP-EhAGCK1 localized to the tunnel part of the membrane (Supplementary Fig. 14) during ingestion of live cells, while GFP-EhAGCK2 showed intense recruitment to the membrane in contact with live Caco-2 cells and followed in the tunnel (Supplementary Fig. 15). As expected, both proteins were absent from newly formed trogosomes. Hence, the results from live cell microscopy indicated that both GFP-EhAGCK1 and -EhAGCK2 showed similar localization pattern during trogocytosis of live CHO and Caco-2 cells.

*E. histolytica* trophozoites ingest dead cells by phagocytosis. The time-lapse imaging with labelled pre-killed CHO cells showed no recruitment of GFP-EhAGCK1 to the phagocytic cups (Fig. 8a and Supplementary Movie 6). The micrographs were further analysed quantitatively, but there was no significant difference in the intensity of GFP-EhAGCK1 at the site of phagocytosis or at newly formed phagosomes compared to plasma membrane and cytosol (Fig. 8b, c). This was confirmed in independent experiments (Supplementary Fig. 16a, b). In contrast, GFP-EhAGCK2 was recruited to the phagocytic site upon contact with dead CHO cells and was later uniformly localized throughout the membrane of phagocytic cups (Fig. 8d and Supplementary Movie 7). Moreover, fluorescence intensity of

GFP-EhAGCK2 showed enrichment in the membrane of the phagocytic cups (Fig. 8e). This was confirmed in independent experiments (Supplementary Fig. 17a, b) Although the protein was present in newly formed phagosomes (Fig. 8f), it eventually started to dissociate from the base of the newly formed phagosomes, following the direction as shown by arrows marked in Supplementary Fig. 18. As observed earlier, the protein dissociated from the newly formed phagosomes rapidly after closure. Overall, our observations suggest that EhAGCK1 is specifically involved in amoebic trogocytosis while EhAGCK2 is involved in all actin-dependent endocytic processes.

## Discussion

Trogocytosis is increasingly recognized as an important process in eukaryotes[24]. It has been observed in many early branching eukaryotes, such as *E. histolytica*, *Naegleria fowleri* and *Hartmanella* as well as in mammals. This suggests that the process has an ancient origin and has evolved early in the evolution of eukaryotes. *N. fowleri* was first shown in 1979 to destroy secondary mouse embryo (ME) cells by a phagocytosis-like mechanism, chewing small portions of ME cytoplasm from the live cell[25]. Since then, trogocytosis has been shown to be involved

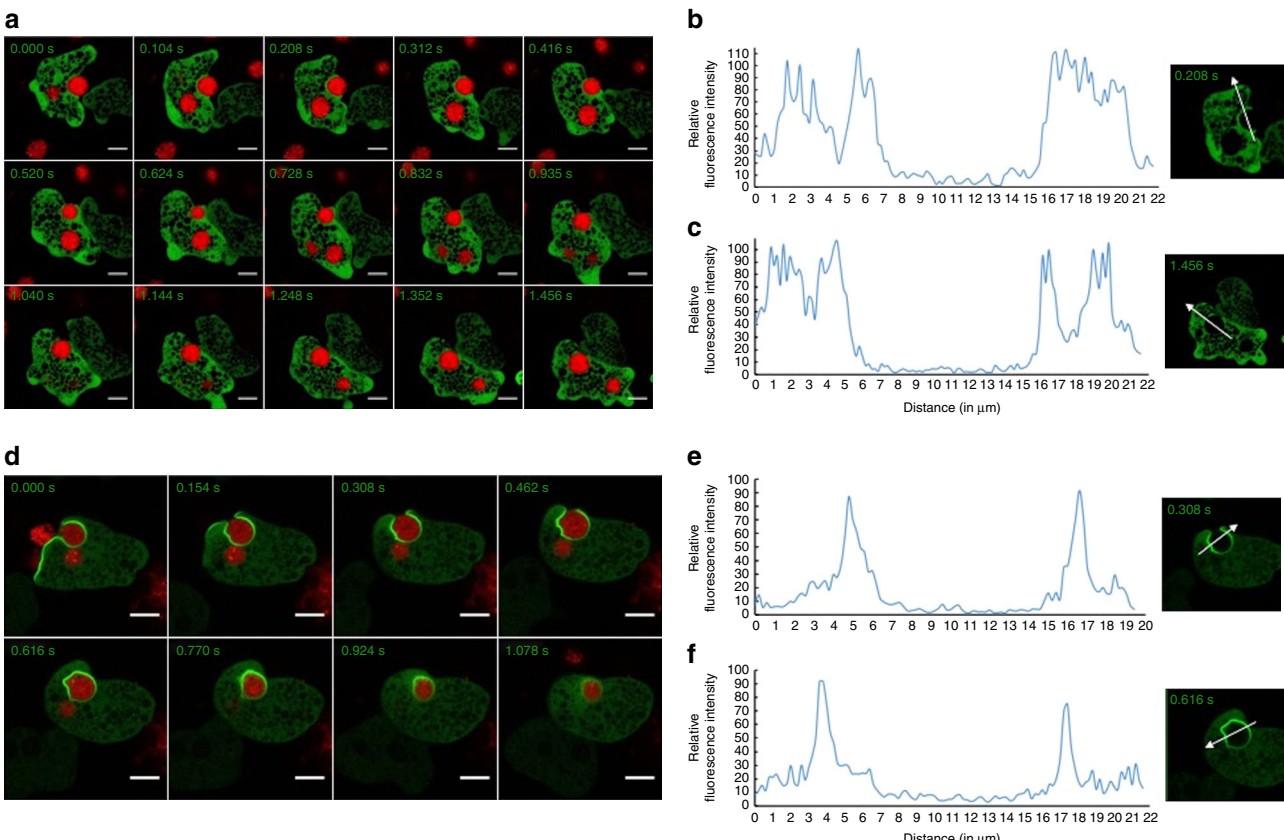

**Fig. 8** Time series montage showing localization of GFP-EhAGCK1 and -EhAGCK2 during phagocytosis of pre-killed CHO cells. **a** Live cell imaging montage showing phagocytosis of pre-killed labelled CHO cells by GFP-EhAGCK1-expressing trophozoites (*Scale bar*, 5 μm). **b**, **c** Plot showing intensity of GFP-EhAGCK1 along the *line* drawn across the phagocytosis cup **b** and newly formed phagosome **c**. **d** Montage showing time series of GFP-EhAGCK2 expressing trophozoites phagocytosing pre-killed labelled CHO cells. **e**, **f** Intensity of GFP-EhAGCK2 along the *line* drawn across phagocytic cup **e** and newly formed phagosome **f**. (*Scale bar*, 10 μm)

in a few biological processes, including acquisition of surface molecules by lymphocytes and cytolysis and tissue invasion by *E. histolytica*[9, 26, 27]. The mechanistic details underlying trogocytosis remain largely unknown; particularly differences in signalling pathways leading to trogocytosis or phagocytosis are unclear. Two small GTPases, TC21 and RhoG, have been reported to participate in T-cell trogocytosis[28]. Moreover, trogocytosis by CD4+ cells involves PI3K, Src and syk kinases[29]. In *E. histolytica*, EhC2PK is thought to be a key molecule in the process[9, 13]. Since all proteins known to be involved in mammalian phagocytosis participate in trogocytosis, it appears that trogocytosis and phagocytosis may be intricately linked. We believe that *E. histolytica* is a good system to study these two processes as the organism displays a high rate of both of these processes and can be easily visualized and quantitated.

We have used trogocytosis of live CHO and Caco-2 cells by *E. histolytica* for identification of the molecules that are present during trogocytosis and absent in phagocytic processes. In order to identify the molecules, we carried out affinity pull down of PtdIns(3,4,5)P$_3$-binding molecules followed by mass spectrometry. This strategy was essentially based on the fact that PtdIns(3,4,5)P$_3$ effectors are present during endocytic processes and dissociate from phagosomes before or soon after the closure. Our search led us to AGC family kinases; EhAGCK1 was found to be associated specifically with trogocytosis, while EhAGCK2 seemed to be involved in all actin-dependent endocytic processes. Our evidence in support of this conclusion is based on gene silencing, quantitative assays and extensive live cell imaging. The EhAGCK1 gene silencing experiments showed a defect in destruction of

live cells specifically, while pinocytosis and phagocytosis of dead cells and RBCs remained unaffected. On the other hand, EhAGCK2-silenced cells showed that it is an essential kinase required for actin-dependent endocytic processes. Furthermore, overexpression of kinase dead versions of EhAGCK1 and EhAGCK2 exhibited a dominant-negative phenotype, verifying their role in destruction of live host cells.

The genome of *E. histolytica* codes for 24 AGC family kinases, none of which have been so far shown to be involved in endocytic processes. Although it has been shown that PI3K-PKC activity is required for host cell killing in *E. histolytica*, further proteins in the pathway have not been identified[30, 31].

The family of AGC kinases is known to influence actin dynamics by operating downstream of PI3K[32, 33]. Akt phosphorylates β-actin in a PtdIns(3,4,5)P$_3$-dependent manner, generally resulting in enhanced cell migration[34]. Moreover, PKC phosphorylates myristoylated alanine rich C-kinase substrate, which plays a crucial role in F-actin cross-linking by integrating the signalling from PKC and calcium-calmodulin[35]. Both Akt and PKC influence actin remodelling in vivo but via different pathways, which implies that actin cytoskeleton is remodelled by different mechanisms during motility and endocytic processes. We hypothesize that actin is also remodelled differently during trogocytosis and phagocytosis. EhAGCK1 may be responsible for relaying the signalling for forming tunnel-like structures during later stages of trogocytosis, while EhAGCK2 may be involved in early stages of actin remodelling, the steps that may be common during all endocytic processes. The dominant-negative phenotype observed in trophozoites overexpressing the kinase dead versions

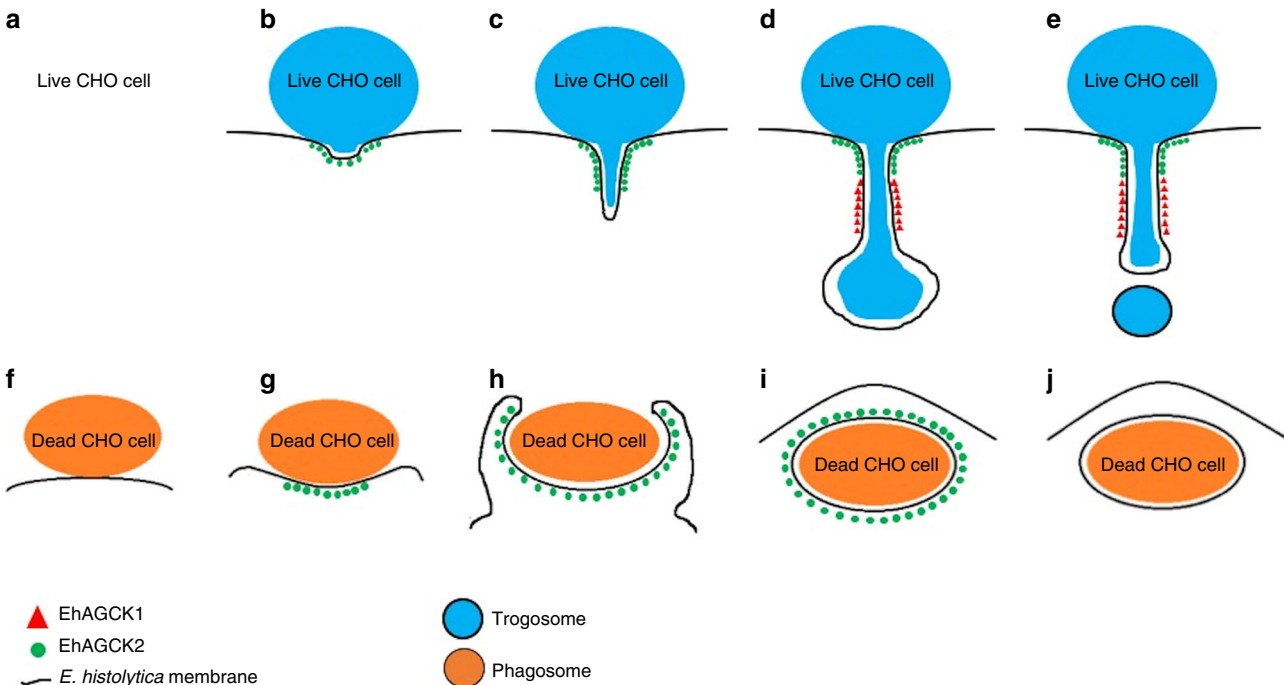

**Fig. 9** Schematic presentation of localization of EhAGCK1 and EhAGCK2 during **a–e** trogocytosis of live CHO cells and **f–j** phagocytosis of dead CHO cells

of EhAGCK1 and EhAGCK2 has indicated the role of kinase activity in the signalling pathway. It is remarkable that both kinases belong to the same family and share 51% identity at amino-acid level but have different roles in amoebic biology. Although both EhAGCK1 and EhAGCK2 show localization to the site of trogocytosis, their pattern of recruitment appears different as depicted schematically in Fig. 9a–e. This difference in localization suggests that both protein kinases relay different signals for remodelling cytoskeleton during trogocytosis or other endocytic processes. It is also worth noting that trogocytosis of live CHO and Caco-2 cells initiated the same pattern of recruitment of EhAGCK1 and EhAGCK2, which indicates that similar signalling pathways are stimulated upon ingestion of different kinds of live host cells. The recruitment of proteins to the site of endocytosis is likely dependent on the PH domain of the proteins, while localization of kinase dead mutants in trophozoites is not affected. The ratio of kinase dead protein to wild-type protein at the site of endocytosis may determine the final fate of the process and could explain the observed dominant-negative phenotype. Our observations indicate that EhAGCK1 is recruited after EhAGCK2, which is similar to the sequential recruitment of PKCε and PKCα during FcγR-mediated phagocytosis[36]. EhAGCK1 may play a role in downstream signalling necessary for formation of initiation complexes, which may include membrane-remodelling molecules such as BAR domain-containing proteins that have been shown to be associated with phagocytosis and virulence[37]. In contrast, in case of phagocytosis of pre-killed CHO cells, only EhAGCK2 is recruited to the phagocytic cups (Fig. 9f–j). Our data also suggest that both EhAGCK1 and EhAGCK2 dissociate from newly formed phagosomes or trogosomes. This is supported by the observation that both EhAGCK1 and EhAGCK2 are absent from previously reported phagosome proteome data sets[38, 39]. Unfortunately, trogosome proteome data are not available. It is not clear how these molecules leave endocytic or trogocytic complexes after progression of the initiation complex towards endosome or trogosome closure.

In conclusion, we have described identification and functional validation of a specific marker of trogocytosis that has allowed us to distinguish trogocytosis from other endocytic processes. In the future, EhAGCK1 could help our understanding of molecular mechanisms of trogocytosis and offer a potential target for development of new drugs.

## Methods

**Organisms and culture.** Trophozoites of *E. histolytica* strain HM-1:IMSS cl-6 (shared by Dr Louis S. Diamond, Laboratory of Parasitic Diseases, NIAID, National Institutes of Health, Bethesda, USA) were cultured axenically at 35 °C in 6 ml screw-capped Pyrex glass tubes or plastic culture flasks in BI-S-33 medium as previously described[40]. For the detection of proteins in the lysates and endocytic assays, ~1×10^6 trophozoites of the late-logarithmic growth phase were cultivated in 15 ml of BI-S-33 medium under anaerobic conditions using Anaerocult A (Merck, Darmstadt, Germany) on a 90 mm culture plate at 35 °C for 2 h. CHO cells (kind gift from Dr Hanada, Department of Biochemistry and Cell Biology, National Institute of Infectious Diseases) were grown in F12 medium (Invitrogen-Gibco) supplemented with 10% foetal bovine serum on a 10-cm-diameter tissue culture dish (IWAKI, Tokyo, Japan) under 5% $CO_2$ at 37 °C. Caco-2 cells (kind gift from Dr Michinaga Ogawa, Department of Bacteriology I, National Institute for Infectious Diseases) were grown in DMEM medium (Invitrogen-Gibco) supplemented with 10% FBS and 1× MEM non-essential amino-acid mix (Sigma-Aldrich).

**Cloning of various constructs used.** The EhAGCK1 (EHI_188930) and EhAGCK2 (EHI_053040) protein-coding region was amplified by PCR from cDNA using specific oligonucleotides containing appropriate restriction sites. A sequence tag consisting of three tandem repeats of the HA peptide was inserted at the amino terminus together with an engineered NheI site[41]. A PCR-amplified DNA fragment was digested with SmaI and XhoI, and ligated into SmaI and XhoI sites of the expression vector pEhEx[41], to produce recombinant plasmid. The HA-tagged kinase dead mutants were generated by mutating the conserved lysine and aspartate residues critical for protein kinase activity by site-directed mutagenesis. The conserved residues K147 and D241 were mutated to alanine in case of EhAGCK1 and K147A and D242A were introduced in case of EhAGCK2. For gene silencing of EhAGCK1 and EhAGCK2, the 420 bp long 5′-end of the protein-coding region was amplified by PCR from cDNA using sense and antisense oligonucleotides. The PCR-amplified DNA fragment was digested with StuI and SacI, and ligated into StuI- and SacI-digested pSAP2-gunma[22]. The gene-silenced strains were established by the transfection of G3 strain (kind gift from Dr David Mirelman, Department of Biological Chemistry, Weizmann Institute of Science, Rehovot, Israel) with the corresponding plasmids as described below.

**Production of *E. histolytica* transformants.** Approximately 10^5 trophozoites were seeded onto 35-mm diameter wells of a six-well culture plate and incubated at 35 °C for 1 h. The LipofectAMINE plasmid DNA complexes were prepared in OPTI-MEM I medium (Life Technologies) supplemented with 5 mg/ml

### Figure legend

▲ EhAGCK1
● EhAGCK2
◠ *E. histolytica* membrane

● Trogosome
● Phagosome

L-cysteine and 1 mg/ml ascorbic acid (transfection medium) and pH 6.8. Transfection medium containing 5 µg of the plasmid was mixed with 10 µl of LipofectAMINE PLUS (Life Technologies) and kept at room temperature for 15 min. This mixture was combined with 20 µg (10 µl) of LipofectAMINE, kept at room temperature for 15 min, diluted with 0.5 ml of transfection medium and added to the seeded trophozoites after removing BI-S-33 medium. The plate was then incubated at 35 °C for 4 h. After incubation with the LipofectAMINE-DNA complex, 70–90% of the trophozoites were viable. The trophozoites were transferred to fresh BI-S-33 medium and further cultivated at 35 °C for 24 h. G418 was then added to the cultures at 1 µg/ml initially and increased to 10 µg/ml gradually by increasing at the rate of 1 µg/ml every 24 h. Finally, transformants were maintained at 10 µg/ml of G418 in medium.

**Reverse transcriptase PCR.** DNase I-treated total RNA (5 µg) was used in the RT reaction using Superscript III (Invitrogen) with random hexamer primer. Annealing was carried out at 65 °C for 5 min, followed by extension at 42 °C for 1 h followed by inactivation at 70 °C for 10 min. An aliquot of 1 µl of this RT mix was used for a regular PCR with specific primers (sequences provided in Supplementary Table 2). PCR was performed with cDNA and amplification conditions were as follows: 94 °C for 5 min; then 25 cycles at 94 °C for 30 s, 52 °C for 1 min and 72 °C for 1 min 30 s; and a final extension at 72 °C for 10 min. No amplicon was observed in the absence of RT enzyme in all RT-PCR experiments.

**Lipid overlay assay.** The total *E. histolytica* cells were lysed in the presence of general phosphatase inhibitors like sodium orthovanadate, ß glycerophosphate and sodium fluoride. The lysate was ultracentrifuged at $100,000 \times g$ for 1 h at 4 °C and supernatant obtained was used for incubation with the membranes spotted with different phospholipids (Echelon) at 4 °C for 1 h. After extensive washing, the membranes were probed with anti-HA antibodies, followed by secondary HRP-labelled anti-mouse IgG antibodies (Invitrogen).

**Identification of PtdInsP$_3$-binding proteins.** To identify PtdIns(3,4,5)P$_3$-interacting proteins by affinity purification, we used PtdIns(3,4,5)P$_3$-conjugated beads (purchased from Echelon). The cells were lysed in presence of general phosphatase inhibitors like sodium orthovanadate, ß glycerophosphate and sodium fluoride. The lysate was ultracentrifuged at $100,000 \times g$ for 1 h at 4 °C and supernatant obtained was used for incubation with PtdIns(3,4,5)P$_3$-conjugated beads. Beads conjugated with PtdIns(4,5)P$_2$ and PtdIns(3)P were used as control and beads (from Echelon) with no lipid conjugates was also included in the experiments. The beads were then mixed with SDS-gel-loading dye and boiled for 5 min. The samples were then resolved on gradient SDS-PAGE gel (5–20%). The proteins were then identified by mass spectrometry.

**Mass spectrometric identification of proteins.** The gel pieces from the band were transferred to a siliconized tube and washed in 200 µl 50% methanol. The gel pieces were dehydrated in acetonitrile, rehydrated in 30 µl of 10 mM dithiothreitol (DTT) in 0.1 M ammonium bicarbonate and reduced at room temperature for 0.5 h. The DTT solution was removed and the sample alkylated in 30 µl 50 mM iodoacetamide in 0.1 M ammonium bicarbonate at room temperature for 0.5 h. The reagent was removed and the gel pieces dehydrated in 100 µl acetonitrile. The acetonitrile was removed and the gel pieces rehydrated in 100 µl 0.1 M ammonium bicarbonate. The pieces were dehydrated in 100 µl acetonitrile, the acetonitrile removed and the pieces completely dried by vacuum centrifugation. The gel pieces were rehydrated in 20 ng/µl trypsin in 50 mM ammonium bicarbonate on ice for 30 min. Any excess enzyme solution was removed and 20 µl 50 mM ammonium bicarbonate was added. The sample was digested overnight at 37 °C and the peptides formed extracted from the polyacrylamide in a 100 µl aliquot of 50% acetonitrile/5% formic acid. This extract was evaporated to 15 µl for MS analysis.

The LC-MS system consisted of a ThermoFisher Velos Orbitrap ETD mass spectrometer system with a Protana nanospray ion source interfaced to a self-packed 8 cm × 75 µm id Phenomenex Jupiter 10 µm C18 reversed-phase capillary column. An aliquot of 7 µl of the extract was injected and the peptides eluted from the column by an acetonitrile/0.1 M acetic acid gradient at a flow rate of 0.5 µl/min over 0.3 h (1.3 h for lanes). The nanospray ion source was operated at 2.5 kV. The digest was analysed using the rapid switching capability of the instrument acquiring a full scan mass spectrum to determine peptide molecular weights followed by product ion spectra (20) to determine amino-acid sequence in sequential scans. This mode of analysis produces ~8000 MS/MS spectra (40,000 for lanes) of ions ranging in abundance over several orders of magnitude.

**Measurement of fluid-phase endocytosis.** To measure fluid-phase endocytosis, $5 \times 10^5$ amoebic transformants were incubated in BI-S-33 medium containing the fluorescent fluid-phase marker RITC dextran (2 mg/ml; Mr = 70,000; Sigma-Aldrich, Japan) at 35 °C for indicated time points. The labelled cells were collected and washed three times with ice-cold PBS. The cell pellets were then suspended in 300 µl of 50 mM Tris-HCl, pH 7.0 containing 1% Triton X-100 and vortexed for 15 s. Fluorescence intensity was measured using fluorometer (F-2500, Hitachi, Japan) at excitation and emission wavelengths of 570 and 610 nm respectively.

**Indirect immunofluorescence.** Cells were transferred to 8-mm round wells on a slide glass, fixed with 3.7% paraformaldehyde and permeabilized with 0.2% Trition X-100/PBS as previously described[37]. The cells were then reacted with anti-HA 16B12 mouse monoclonal antibody (1:1000) (Biolegend, 901513). The samples were then reacted with Alexa Fluor 568-conjugated anti-mouse secondary antibody (1:1000) for 1 h and FITC-labelled phalloidin (Sigma, P5282). The samples were examined on a Carl-Zeiss LSM780 confocal laser-scanning microscope. Images were further analysed using LSM780 software.

**Live cell imaging.** Approximately $5 \times 10^5$ transformants were cultured on a 35 mm collagen-coated glass-bottom culture dish (MatTek Corporation, Ashland, MA) in 3 ml of BI-S-33 medium under anaerobic conditions. CHO cells were stained for 30 min with 20 mM CellTracker blue dye or CellTracker orange dye (Molecular probes, Eugene, OR) in F12 medium containing 10% FCS. After staining, CHO cells were washed three times with fresh F12 medium, and ~$2 \times 10^5$ CHO cells in 200 µl F12 medium were added to the GFP-tagged protein-expressing amoeba in a glass-bottom dish. The culture was carefully covered with a coverslip, and overloaded medium was removed. The junction of the coverslip and slide glass was sealed with nail polish, and the culture was incubated at 35 °C in a temperature control unit on Zeiss, LSM780 equipped with a ×63/1.4 oil immersion objective and CCD camera.

**Erythrophagocytosis assay.** Briefly, $10^6$ RBCs were washed with PBS and incomplete BI-S-33 and were then incubated with $10^5$ entamoeba for varying times at 37 °C in 0.5 ml of culture medium. The trophozoites and erythrocytes were centrifuged to get a pellet, non-engulfed RBCs were lysed with cold distilled water and recentrifuged at $1000 \times g$ for 2 min and step was repeated twice, followed by resuspension in 1 ml formic acid to lyse entamoebae containing engulfed RBCs. The absorbance was measured at 400 nm.

**Assay for destruction of CHO and Caco-2 cell monolayers.** The destruction of CHO monolayers was quantified as described previously with slight modifications[23]. Briefly, CHO cells were labelled with 40 mM of CellTracker blue in the growth medium at 37 °C for 2 h. After the medium was replaced with pre-warmed OPTI-MEM (Invitrogen-Gibco) supplemented with 137 mM L-cysteine and 19 mM ascorbic acid, pH 6.7, ~$1.0 \times 10^5$ *E. histolytica* trophozoites were added and incubated at 35 °C for 60 min. In case of Caco-2 destruction, $2.5 \times 10^5$ trophozoites were taken and incubated for 90 min. After this incubation, the remaining CHO/Caco-2 cells were collected using trypsin and the fluorescence of CellTracker blue was measured using a fluorometer (F-2500, Hitachi, Japan) with excitation and emission at 353 and 465 nm, respectively. The number of adherent CHO/Caco-2 cells was proportional to the intensity of CellTracker blue staining and expressed as a percentage of the remaining fluorescence of untreated CHO/Caco-2 cells.

**Trophozoite migration and transwell migration.** Briefly, amoebae were grown in the presence of 10 mg/ml G418 for 24 h, harvested in log-growth phase, suspended in serum-free BI growth medium and $10^5$ cells loaded in the upper chamber of a transwell migration chamber (Costar, 8 mm pore size). Before, transwell porous filter was layered with 0.5% agarose and 12% porcine stomach mucine, and allowed to gel for 2 h prior to assay initiation. The lower chamber contained growth medium with 15% adult bovine serum. Incubation time was 16 h to allow penetration of amoeba through the mucine. Migrated trophozoites attached to the lower chamber wall were detached on ice, fixed and counted. Each experiment was performed in triplicate and statistical significance among three independent experiments was determined by an unpaired, two-tailed Student's *t*-test.

**Statistical analysis.** The statistical analysis of the data was done by using GraphPad software. The statistical comparisons were performed by using unpaired Student's *t*-test.

**Data availability.** The authors declare that the data supporting the findings of this study are available within the article and its Supplementary Information files, or are available from the authors on request.

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

## Acknowledgements

We acknowledge Japanese Society for Promotion of Sciences (JSPS) for providing JSPS postdoctoral fellowship for foreign researchers. This work was supported in part by Grants-in-Aid from JSPS (26293093), Grants-in-Aid for Scientific Research on Innovative Areas (23117001 and 23117005) from the Ministry of Education, Culture, Sports, Science and Technology (MEXT) and a grant for Research on Emerging and Re-emerging Infectious Diseases from the Japan Agency for Medical Research and Development (AMED) to T.N. We thank the Department of Sciences and Technology, India for Inspire Faculty Award. We are also grateful to Prof. Alok Bhattacharya for critically inspecting the manuscript.

## Author contributions

Somlata and T.N. conceived and designed the experiments. Somlata performed the experiments. S. and K.N.-T. analysed the data. T.N. and K.N.-T. contributed reagents, materials and analysis tools. All authors wrote the paper.

## Additional information

**Competing interests:** The authors declare no competing financial interests.

