## [Peer Review File · Nature Communications]

Reviewers' comments:

Reviewer #1 (Remarks to the Author):

This report written by Somlata et al. provides new and important information about the process of trogocytosis in *Entamoeba histolytica*. Given the importance of endocytic mechanisms to the parasite's pathogenicity and the novelty of trogocytosis, the data are significant, timely and unique. This reviewer has the following comments:

1. The first experiment is to identify PIP3-binding proteins by affinity methods. There are no details of these experiments given anywhere. For example, the authors mention that amoebic lysates were cleared by ultracentrifugation; no speeds or times are given. The authors mention that the lysates were incubated with beads; no times or temperatures are given. How was mass spec carried out? Many other details are missing.
2. To assess the function of the two kinases, their expression was knocked down by an antisense-RNA-based method. The authors show successful knockdown by RT-PCR. How do the authors know that EhAGCK1 is not knocked down in the EhAGCK2-knockdown cell line and vice versa? In other words, an important control is missing—to show that you have not knocked down both kinases in one or both of the cell lines. This reviewer realizes that the phenotypes of the two cell lines are different, but it is conceivable, given homology between the two genes, that in the most severe phenotypes both kinases might be knocked down.
3. Related to Comment 2 (above), there are no details given on how the RT-PCR was performed.
4. The authors use HA-tagging to look at the localization of the two kinases in cell (Fig 3). What is the control for non-specific binding of the anti-HA antibody? Do the authors have a cell line that expresses a HA-tagged irrelevant protein?
5. The expression of the HA-tagged transgenes are driven by a constitutive promoter; therefore, there is an over-abundance of the kinases in these cell lines. Has the rate of any of the endocytic processes changed (increased?) in these cell lines? If so, this could confound the interpretation of the localization data.
6. In the Materials and Methods (Lines 429-43), the authors state that, "...gene-silenced strains were established...as described above." However, the description for making transformants follows this statement (i.e., "below").
7. In the Materials and Methods (Lines 443-444) the authors indicate that they select for transformed cells with G418 which was "increased subsequently." Increased to what? Over what time frame?
8. In Materials and Methods (Statistical Analysis; Lines 488-489). What does this sentence mean? "Mostly analysis was performed by using student t-test until mentioned." I think the authors mean they used t-tests unless otherwise noted, but there was no mention of any other statistical test in the report.
9. Related to Comment 8 above, the authors should standardize the statistical significance symbols in the figures (e.g., Figure 2). For example, the authors should stick to one, two or three asterisks. Using various combinations of symbols to denote significance is confusing.
10. The manuscript could have benefited from editing. There are numerous grammar and syntax errors throughout. Below are few examples, but there are many others:

- a. The sentence beginning in line 89 ("It appears that amoeba trogocytosis...") and ending in line 91 ("...different endocytic processes.") is too long to be comprehensible.
- b. The paragraph that spans Lines 145- 150 seems out of place. It seems that it should be somewhere at the end of an abstract or introduction or the beginning of a discussion.
- c. Line 315: Fluorescent intensity should be by fluorescence intensity.

Reviewer #2 (Remarks to the Author):

Review: AGC Family Kinase1 Participates in Trogocytosis but not in Phagocytosis in *Entamoeba histolytica*

Summary: Somlata et al. sought to understand the role of AGC Family Kinase1 (EhAGCK1) in *E. histolytica*. The authors focused in particular on EhAGCK1 because it had been identified while looking for effectors of PtdIns(3,4,5)P₃, which are known to play a role in endocytosis and motility in mammalian systems. The authors knocked down expression of both EhAGCK1 and a second AGC family kinase, EhAGCK2. They found that the EhAGCK1 knockdown did not affect pinocytosis or red blood cell phagocytosis, but did decrease CHO cell killing, which the authors interpret as a decrease in trogocytosis. In contrast, the EhAGCK2 knockdown resulted in defects in all three assays. The authors then investigate the localization of EhAGCK1 and EhAGCK2 during pinocytosis, phagocytosis, trogocytosis using parasites that express HA- or GFP-tagged constructs. The authors report that EhAGCK1 was associated more often with pseudopods, while EhAGCK2 was associated more often with membrane and endocytic structures. During phagocytosis the authors observed increased EhAGCK2 staining at the phagocytic cup. During trogocytosis the authors again found EhAGCK2-GFP fluorescence at the cup, they also observed EhAGCK1-GFP fluorescence at the "neck" of the engulfment site. From these data the authors conclude that EhAGCK1 plays a role in trogocytosis, but not phagocytosis or pinocytosis. Overall, the paper is well-written, state of the art, represent an important advance, and appropriate controls were used.

- Fig 2

o Based on these data the authors write that for EhAGCK1 "silencing led to a defect in the CHO cells destruction and trogocytosis" but not pinocytosis or phagocytosis. However, they fail to assess trogocytosis directly in the knockdown parasites. It would therefore be more appropriate to leave out the word trogocytosis from the abstract.

o The erythrophagocytosis data is striking, but difficult to evaluate because the methods were not adequately explained either in the figure legend or the methods section.

- Fig 4

o The authors demonstrate GFP-EhAGCK1 signal in the pseudopods, suggesting a potential role in motility, however the motility of the EhAGCK1 knockdown strain is not addressed. The authors use destruction of a CHO cell monolayer as a surrogate for trogocytosis. However, this assay requires the amoebae to be motile and move around the monolayer to

kill multiple cells. It is possible that a parasite with a motility defect might still undergo trogocytosis, but might exhibit a cell killing defect due to its impaired ability to reach as many CHO cells. The authors should address the motility of the EhAGCK1 knockdown parasites.

Reviewer #3 (Remarks to the Author):

In this manuscript, Dr Nozaki and colleagues identified a kinase from *Entamoeba histolytica*, namely EhAGCK1, specifically involved in trogocytosis but not related processes such as (erythro)-phagocytosis or pinocytosis. This is the main claim of the study and it is a novel and important one since tools capable of dissecting trogocytosis from other related processes are clearly lacking, which limits the investigation of the functional consequences of trogocytosis.

PI3Kinase being a key molecule common to endocytosis related processes such as pinocytosis, phagocytosis or trogocytosis, the authors made use of PtdIns(3,4,5)P-coated beads to fish out specific molecular partners in amoeba lysates and screened for candidate proteins specifically involved in trogocytosis. This strategy led to the discovery of EhAGCK1 as a candidate fulfilling the expected requirements. Amoebae silenced for EhAGCK1 expression were impaired in their ability to kill target cells. Because trogocytosis and phagocytosis are both involved in the pathogenesis of amoebiasis, the identification of kinases specifically involved in one of these processes could provide a powerful tool to study the mechanisms behind the pathogenesis of amoebiasis and this kinase might represent a target for the design of specific drugs. Thus, the topic of the study is of importance and the success in identifying EhAGCK1 opens new ways for future investigations.

Trogocytosis is the name given to the process whereby various types of cells (including mammalian cells of the immune system and amoeba) capture portion of the cells they interact with. Trogocytosis therefore appears as a conserved process but the mechanisms might be slightly different depending on the type of cells involved. The kinase identified here belongs to the world of amoeba precluding a direct translation to the trogocytosis process involved in mammalian cells but the study is nevertheless of a large interest in many areas of investigation and suggests that the identification of mammalian molecules selectively involved in trogocytosis can be a goal possible to achieve. In that, this study has a good potential to stimulate thinking in the field of amoeba and beyond.

Overall the identification of the amoeba-derived PtdIns(3,4,5)P interacting proteins is well done, most of the images and videos support the conclusions drawn by the authors and the differential contribution of EhAGCK1 and 2 in the various endocytosis-related processes analysed by the authors is also convincing.

However, the paper is currently mostly descriptive and additional experiments are required to strengthen the author's conclusions and to make sure the identified kinase is indeed a selective effector of trogocytosis. Please find below my detailed comments:

Major comments

1) In line 145 (page 5 of the manuscript), the authors indicate that they « screened » proteins for their roles in various processes. It is unclear whether this screening step is equivalent or differs from the results presented in Fig. 2. Because the screening step is used by the authors to narrow down the list of 18 candidates to only one (EhAGCK1) plus a control (EhAGCK2), it is indispensable to make it clear the description of the experimental procedures for the (apparently) initial screening.

2) Figure 2c. CHO cell killing is used here as surrogate trogocytosis assay. Although trogocytosis has been proposed to reflect the initial mechanism of killing, a direct analysis of trogocytosis, i. e. capture of CHO target cell portions by amoeba should be performed since the mechanisms linking trogocytosis and target cell killing still remain to be fully established (and cannot be considered as strictly and uniquely related processes at the moment). The fact that tagged-EhAGCK1 localizes specifically to the trogocytosis sites (as shown in Fig. 6) is not a direct proof for the involvement of EhAGCK1 in trogocytosis.

3) Figure 2b. It is unclear to this reviewer if the erythrophagocytosis assay involves an initial step of trogocytosis of red blood cells before phagocytosis ensues (as shown by Ralston et al. the authors' reference #9). If so, why is this assay insensitive to EhAGCK1 silencing ?

4) It would also be important to analyse trogocytosis of more relevant cells (such as epithelial cells) by *E. histolytica* and its EhAGCK1 (or 2)-silenced counterpart. If the authors master the tissue invasion assay reported in their reference #9, it would also be fascinating and very important to analyse the impact of silencing EhAGCK1 on tissue invasion by *E. histolytica*.

Minor comments

5) The manuscript needs rewriting by people whose english is their mother tongue. In addition, the materials and methods section needs to be more carefully written. For instance, there are no references to the PtdIns(3,4,5)P3 conjugated beads : are they from commercial source (if so, please provide the origin) or homemade (if so, please provide appropriate information and controls to show the specificity for existing PtdIns(3,4,5)P3 partners)?

6) Figure 1b is described in the legend as c) and vice versa.

7) Could the authors quantify the reduction in kinase expression in silenced cells (by RTqPCR for instance)?

8) Figures dealing with microscopy require quantitative and statistical analysis analysis (very often one or two cells only are shown).

General Response to reviewers

We are grateful to the reviewers for their thoughtful critiques. In the revised work, we have addressed all of the comments, through new experimentation or clarification of details in the text. We have performed new experiments to address the points raised by the reviewers.

We sincerely thank the reviewers for all of their comments, as we truly believe that the edits and experiments we performed in response to the critiques have improved the manuscript. Please find below a detailed point-by-point response to the comments raised by each reviewer.

Response to Reviewer #1

1. The first experiment is to identify PIP3-binding proteins by affinity methods. There are no details of these experiments given anywhere. For example, the authors mention that amoebic lysates were cleared by ultracentrifugation; no speeds or times are given. The authors mention that the lysates were incubated with beads; no times or temperatures are given. How was mass spec carried out? Many other details are missing.

Response: We agree with the reviewers comment and we have provided the protocols for affinity based isolation of PtdIns(3,4,5)P₃ binding proteins, mass spectrometric technique and erythrophagocytosis in supplementary information.

2. To assess the function of the two kinases, their expression was knocked down by an antisense-RNA-based method. The authors show successful knockdown by RT-PCR. How do the authors know that EhAGCK1 is not knocked down in the EhAGCK2-knockdown cell line and vice versa? In other words, an important control is missing—to show that you have not knocked down both kinases in one or both of the cell lines. This reviewer realizes that the phenotypes of the two cell lines are different, but it is conceivable, given homology between the two genes, that in the most severe phenotypes both kinases might be knocked down.

Response: We appreciate the critical comment of the reviewer and analyzed the expression of EhAGCK1 in EhAGCK2 gene silenced cell lines and vice versa (Supplementary Fig. S4). Our results show that knockdown was very specific for the respective genes.

3. Related to Comment 2 (above), there are no details given on how the RT-PCR was performed.

Response: We agree with the reviewer's comment and have provided the protocol in supplementary information.

4. The authors use HA-tagging to look at the localization of the two kinases in cell (Fig 3). What is the control for non-specific binding of the anti-HA antibody? Do the authors have a cell line that expresses a HA-tagged irrelevant protein?

*Response: In agreement with reviewer's comment we have provided the data for HA antibody control and HA-tagged mock protein expressed in *E. histolytica* trophozoite under same conditions in Supplementary Fig. S8c and 8d. The HA-tagged mock protein neither localizes to phagocytic cup (marked with arrow) nor to the newly formed phagosome (marked with star). HA-tag does not influence the localization of protein in trophozoites.*

5. The expression of the HA-tagged transgenes are driven by a constitutive promoter; therefore, there is an over-abundance of the kinases in these cell lines. Has the rate of any of the endocytic processes changed (increased?) in these cell lines? If so, this could confound the interpretation of the localization data.

Response: We thank reviewer for critical comment regarding overexpression of EhAGCK1 and 2 under constitutive expression system. We analyzed the pinocytosis and erythrophagocytosis by the trophozoites overexpressing the HA-tagged gene in comparison to vector control (supplementary fig. S5a and b). Our results indicate no significant increase in pinocytic or erythrophagocytic capability in overexpressing trophozoites as compared to control trophozoites. This has been previously observed in case of EhC2PK, where overexpression does not lead to increase in erythrophagocytosis (Ref no. 13).

6. In the Materials and Methods (Lines 429-43), the authors state that, "...gene-silenced strains were established...as described above." However, the description for making transformants follows this statement (i.e., "below").

Response: We agree with reviewer and corrected the typographical error.

7. In the Materials and Methods (Lines 443-444) the authors indicate that they select for transformed cells with G418 which was "increased subsequently." Increased to what? Over what time frame?

Response: We have added the details in the protocol to address reviewer's concern.

8. In Materials and Methods (Statistical Analysis; Lines 488-489). What does this sentence mean? "Mostly analysis was performed by using student t-test until mentioned." I think the authors mean they used t-tests unless otherwise noted, but there was no mention of any other statistical test in the report.

Response: We thank reviewer for critical reading and corrected the typographical error. All the statistical analysis has been done by unpaired, student's t-test.

9. Related to Comment 8 above, the authors should standardize the statistical significance symbols in the figures (e.g., Figure 2). For example, the authors should stick to one, two or three asterisks. Using various combinations of symbols to denote significance is confusing.

Response: In response to reviewer's concern this has been corrected and followed throughout the manuscript.

10. The manuscript could have benefited from editing. There are numerous grammar and syntax errors throughout. Below are few examples, but there are many others:
a. The sentence beginning in line 89 ("It appears that amoeba trogocytosis...") and ending in line 91 ("...different endocytic processes.") is too long to be comprehensible.

Response: We thank reviewer for comment and in agreement with reviewer's concern we have re written the parts of manuscript and tried to avoid grammatical mistakes as much as possible. We will avail journal's language editing services.

b. The paragraph that spans Lines 145-150 seems out of place. It seems that it should be somewhere at the end of an abstract or introduction or the beginning of a discussion.

Response: We appreciate reviewer's opinion but in our understanding, the identification of PtdInsP₃ proteins is the suitable heading for introducing EhAGCK1 and 2 in manuscript.

c. Line 315: Fluorescent intensity should by fluorescence intensity.

Response: In response to reviewer's comment, we have corrected the typographical error.

Response to Reviewer #2:

1. Based on these data the authors write that for EhAGCK1 "silencing led to a defect in the CHO cells destruction and trogocytosis" but not pinocytosis or phagocytosis. However, they fail to assess trogocytosis directly in the knockdown parasites. It would therefore be more appropriate to leave out the word trogocytosis from the abstract.

Response: We thank reviewer for the critical comment. We focused on evaluating the defect in internalization of fragments of live Caco-2 cells which are relatively flat and large. Since live amoeba are highly motile it was difficult to quantitatively analyze the ingested fragments in real time. For quantitative evaluation, we incubated the gene silenced trophozoites with labelled live Caco-2 cells for 30 min and then fixed the amoebic cells. The cells were then analyzed by microscopy. The fields and cells were randomly selected and cells were z-sectioned. The images were reconstructed in IMARIS 7.6 software and individual amoebic cells were then analyzed for total intensity of labelled Caco-2 cell fragments ingested. In each experiment 45 cells were selected and it was repeated 3 times. We could see statistically significant difference in average intensity of labeled Caco-2 cell fragments ingested by the gene silenced trophozoites as compared the vector control. This data has been included as supplementary fig. S7. The reduction in the fragments of live caco-2 cell ingested by EhAGCK1gs and EhAGCK2gs cells indicate defect in trogocytosis by the parasite.

2. The erythrophagocytosis data is striking, but difficult to evaluate because the methods were not adequately explained either in the figure legend or the methods section.

Response: We thank reviewer and have included the protocol in supplementary information.

3. The authors demonstrate GFP-EhAGCK1 signal in the pseudopods, suggesting a potential role in motility, however the motility of the EhAGCK1 knockdown strain is not addressed. The authors use destruction of a CHO cell monolayer as a surrogate for trogocytosis. However, this assay requires the amoebae to be motile and move around the monolayer to kill multiple cells. It is possible that a parasite with a motility defect might still undergo trogocytosis, but might exhibit a cell killing defect due to it's impaired ability to reach as many CHO cells. The authors should address the motility of the EhAGCK1 knockdown parasites.

Response: We appreciate the critical comment of the reviewer. In the CHO cell destruction assay we have used confluent monolayer of the mammalian cells and it is expected that motility of parasite should not be limiting factor for cell destruction. But to address reviewer's concern we performed trans-well migration assay to

assess the motility of parasites. The assay indicated 20% defect in the motility of parasites silenced for EhAGCK1. Although the p value for the experiment was significant but the defect was less prominent. In the experimental situation, where the mammalian cells are not limiting we assume that 20% defect in motility will not affect the results. Furthermore, this may be of importance in *in vivo* conditions but it needs to be investigated in tissue models and animal models. In Our understanding 20% defect in motility may not significantly affect the cell killing in experimental situation.

Response to Reviewer #3

Major comments

1) In line 145 (page 5 of the manuscript), the authors indicate that they « screened » proteins for their roles in various processes. It is unclear whether this screening step is equivalent or differs from the results presented in Fig. 2. Because the screening step is used by the authors to narrow down the list of 18 candidates to only one (EhAGCK1) plus a control (EhAGCK2), it is indispensable to make it clear the description of the experimental procedures for the (apparently) initial screening.

Response: We thank reviewer for critical reading and comment. The initial screening procedure involved tagging the proteins at N-terminal with HA tag and observing the localization of the tagged proteins. The data was further cross examined by analyzing the live cell imaging of GFP-tagged proteins. On the basis of microscopy data, proteins selected were then selected for gene silencing and further experiments were carried out. We have included the strategy adopted for screening in form of a flow diagram in supplementary fig. S1.

2) Figure 2c. CHO cell killing is used here as surrogate trophocytosis assay. Although trophocytosis has been proposed to reflect the initial mechanism of killing, a direct analysis of trophocytosis, i. e. capture of CHO target cell portions by amoeba should be performed since the mechanisms linking trophocytosis and target cell killing still remain to be fully established (and cannot be considered as strictly and uniquely related processes at the moment). The fact that tagged-EhAGCK1 localizes specifically to the trophocytosis sites (as shown in Fig. 6) is not a direct proof for the involvement of EhAGCK1 in trophocytosis.

Response: We thank reviewer for the critical comment. To establish specific relation between EhAGCK1 or 2 and destruction of live host cells, we overexpressed the kinase dead mutant of the genes and analyzed the defect in time dependent manner for CHO and Caco-2 cells. We observed the dominant negative phenotype due to overexpression of kinase dead mutants, the data has been included in supplementary fig. S6b and c. The data obtained from both gene silencing and kinase dead mutant overexpression experiments indicate EhAGCK1 and 2 are involved in destruction of live mammalian cells. For analyzing the fragments of Live cells ingested by trophozoites, we used microscopy of gene silenced cells fed with live Caco-2 cells. The 3-d reconstructed images show reduction in fragments of live cells ingested by trophozoites as compared to vector control. We also quantitated the data for 135 cells per transfected cell line (45 cells from each experiment per cell line) by IMARIS7.6 software. The data has been shown in supplementary fig. S7.

3) Figure 2b. It is unclear to this reviewer if the erythrophagocytosis assay involves an initial step of trophocytosis of red blood cells before phagocytosis ensues (as shown by Ralston et al. the authors' reference #9). If so, why is this assay insensitive to EhAGCK1 silencing?

Response: We appreciate the critical comment of the reviewer. We do agree that erythrocytes are also internalized by trophocytosis by *E. histolytica* trophozoites as published previously. But in our understanding *Entamoeba* prefers to ingest large and flat live host cells of gut epithelium and liver through trophocytosis as it is an energetically efficient process. While due to small size of erythrocytes they can be easily internalized through phagocytosis also as compared to large flat live host cells. In our understanding, *E. histolytica* trophozoites when silenced for EhAGCK1 gene may have switched to phagocytosis of erythrocytes in order to adapt the stress of gene silencing. It is also well known that *Entamoeba* can be highly flexible in terms of signaling pathways in order to survive in host system. Hence, we speculate that silencing of EhAGCK1 does not lead to significant effect on erythrophagocytosis by the trophozoites.

4) It would also be important to analyze trophocytosis of more relevant cells (such as epithelial cells) by *E. histolytica* and its EhAGCK1 (or 2)-silenced counterparts. If the authors master the tissue invasion assay reported in their reference #9, it would also be fascinating and very important to analyze the impact of silencing EhAGCK1 on tissue invasion by *E. histolytica*.

Response: We thank reviewer for the comment. We have included the data for live cell imaging of GFP-EhAGCK1 or 2 and live Caco-2 cells in the manuscript (Supplementary fig. S10 and S11). The localization pattern of both GFP-EhAGCK1 and 2 during trophocytosis of Caco-2 cells is similar to previously observed for CHO cells. Also, we included data for Caco-2 monolayer destruction by gene silenced and trophozoites overexpressing kinase dead versions of the gene, the data is presented in Fig. 2c and supplementary fig. S6b respectively. We are grateful for reviewer's suggestion of analyzing tissue invasion by gene silenced trophozoites but in current time frame it was not possible for us to perform the experiment but we are working on developing expertise on the technique for future studies.

Minor comments

5) The manuscript needs rewriting by people whose English is their mother tongue. In addition, the materials and methods section needs to be more carefully written. For instance, there are no references to the PtdIns(3,4,5)P₃ conjugated beads: are they from commercial source (if so, please provide the origin) or homemade (if so, please provide appropriate information and controls to show the specificity for existing PtdIns(3,4,5)P₃ partners)?

Response: We agree with reviewer's concern. We have provided the detailed protocol including the source of reagents. Our Fig. 1c includes the SDS-PAGE profile for the proteins which eluted from PtdIns(3,4,5)P₃ resin and other control beads. The mass spectrometry data was further analyzed and compared with control experiments to identify proteins which were uniquely present in PtdIns(3,4,5)P₃ data.

6) Figure 1b is described in the legend as c) and vice versa.

Response: We have corrected the typographical error and thank reviewer for the

comment.

7) Could the authors quantify the reduction in kinase expression in silenced cells (by RTqPCR for instance)?

Response: We thank reviewer for the critical comment. We have repeated knock down experiments and we get 100% silencing of the genes. Since our Reverse Transcriptase-PCR results for 25 cycles show complete silencing we believe the gene has been silenced in the trophozoites.

8) Figures dealing with microscopy require quantitative and statistical analysis (very often one or two cells only are shown).

Response: We agree with reviewer's concern. The figures presented in the manuscript are representative and have been analyzed for fluorescence intensities of GFP-tagged proteins individually. But we would like to highlight the fact that tunnels formed during trogocytosis are highly dynamic and keep shifting the planes during microscopy. In data Fig. 7a, first two frames have another trogocytic tunnel formed which is out of plane. So, it becomes very difficult to capture multiple events in a single field. But we will be happy to provide more microscopy results if needed as supplementary data.

Reviewers' Comments:

Reviewer #1 Remarks to the Author:

This report written by Somlata et al. provides an important advance in our understanding of the process of trophocytosis in *Entamoeba histolytica*. Trophocytosis contributes to virulence. Therefore, the findings are significant. The authors have adequately responded to the reviewers' critiques. This reviewer has the following minor comments:

1. In the introduction (Lines 61-67), the authors describe the situation whereby only a fraction of those infected with *E. histolytica* exhibit symptoms. They call parasites that don't cause symptoms avirulent and those that cause symptoms virulent. They also mention a switch from avirulence to virulence. This is not precise. I think the authors mean invasive and non-invasive.
2. In two places the authors use strong language that is not conventional. For example, in Line 215 the authors state that, "...results were further proven..." and in Line 298 the authors state, "...to further confirm..." Nothing in science is proven or confirmed. Rather, findings can be "supported" or "refuted."
3. In Fig. 1C, the arrows below the phosphorylation sites are not defined. In the legend, the authors mention that the conserved phosphorylation sites are shown, but they don't say shown by an arrow.

Reviewer #2 Remarks to the Author:

The authors have comprehensively responded to the critiques and the manuscript is improved as a result.

Reviewer #3 Remarks to the Author:

In their revised manuscript, Dr Nozaki and colleagues have addressed most of the comments raised by all three reviewers, which resulted in multiple changes in the text as well as the provision of new data. Overall, the manuscript is greatly improved and in particular the addition of a set of data reporting directly on the process of trophocytosis (instead of a killing assay) is very helpful and is a big improvement. The manuscript still suffers from the lack of an in vivo experiment assessing tissue invasion by *Entamoeba histolytica* silenced for EhAGCK1 expression but I understand this is very technically challenging and that the authors are not in a position to perform this experiment. As it is, I believe the study will be of interest for a broad audience of scientists including but not limited to microbiologists and cellular biologists.

I would recommend the authors to :

- i) refer to the first description of trophocytosis performed by Brown in 1979 (J Med Microbiol. 1979 Aug;12(3):363-71.)
- ii) provide more microscopy results to compensate for the lack of statistical analysis of the microscopy data
- iii) edit the manuscript with the help of a native English writer

Reviewer #1 (Remarks to the Author):

1. In the introduction (Lines 61-67), the authors describe the situation whereby only a fraction of those infected with *E. histolytica* exhibit symptoms. They call parasites that don't cause symptoms avirulent and those that cause symptoms virulent. They also mention a switch from avirulence to virulence. This is not precise. I think the authors mean invasive and non-invasive.

Response: We thank reviewer for the critical reading. We have suitably changed the text.

2. In two places the authors use strong language that is not conventional. For example, in Line 215 the authors state that, "...results were further proven..." and in Line 298 the authors state, "...to further confirm..." Nothing in science is proven or confirmed. Rather, findings can be "supported" or "refuted."

Response: We thank reviewer for the comment and have changed the text suitably.

3. In Fig. 1C, the arrows below the phosphorylation sites are not defined. In the legend, the authors mention that the conserved phosphorylation sites are shown, but they don't say shown by an arrow.

Response: We have defined the arrows in the legend as per suggestion of the reviewer.

Reviewer #3 (Remarks to the Author):

i) refer to the first description of trogocytosis performed by Brown in 1979 (J Med Microbiol. 1979 Aug;12(3):363-71.)

Response: We have added the reference as per reviewer's suggestion. We thank reviewer for the reference provided.

ii) provide more microscopy results to compensate for the lack of statistical analysis of the microscopy data

Response: We have included more microscopic data in supplementary data section for compensating the lack of statistical analysis. We thank reviewer for understanding our work and limitations.

iii) edit the manuscript with the help of a native english writer

Response: We will be surely availing language editing service provided by Nature publications. We thank reviewer for the suggestion.